# Comparison of two methods to assess physical activity prevalence in children: an observational study using a nationally representative sample of Scottish children aged 10–11 years

Paul McCrorie, Rich Mitchell, Anne Ellaway

MRC/CSO Social & Public Health Sciences Unit, University of Glasgow, Glasgow, UK

**Correspondence to**
Dr Paul McCrorie;
paul.mccrorie@glasgow.ac.uk

## ABSTRACT

**Objectives** To describe the objectively measured levels of physical activity (PA) and sedentary time in a nationally representative sample of 10–11-year-old children, and compare adherence estimates to the UK PA guidelines using two approaches to assessing prevalence.

**Design** Nationally representative longitudinal cohort study.

**Setting** Scotland wide in partnership with the Growing up in Scotland (GUS) study. Data collection took place between May 2015 and May 2016.

**Participants** The parents of 2402 GUS children were approached and 2162 consented to contact. Consenting children (n=1096) wore accelerometers for 8 consecutive days and 774 participants (427 girls, 357 boys) met inclusion criteria.

**Primary and secondary outcome measures** Total PA (counts per minute, cpm); time spent sedentary and in moderate-to-vigorous PA (MVPA); proportion of children with ≥60 min MVPA on each day of wear (daily approach); proportion of children with ≥60 min of MVPA on average across days of wear (average approach)—presented across boys and girls, index of multiple deprivation and season.

**Results** Mean PA level was 648 cpm (95% CI, 627 to 670). Children spent 7.5 hours (7.4–7.6) sedentary/day and 72.6 min (70.0–75.3) in MVPA/day. 11% (daily) and 68% (average) of children achieved the recommended levels of PA (P<0.05 for difference); a greater proportion of boys met the guidelines (12.5% vs 9.8%, NS; 75.9% vs 59.5%, P<0.001); guideline prevalence exhibited seasonal variation. No significant socioeconomic patterning existed across any outcome measure.

**Conclusions** PA estimates are significantly influenced by the analytical method used to assess prevalence. This could have a substantial impact on the evaluation of interventions, policy objectives and public health investment. Existing guidelines, which focus on daily PA only may not further our understandings about the underlying construct itself. Gender differences exist within this age-group, suggesting greater investment, with particular consideration of seasonality, is needed for targeted intervention work in younger children.

## Strengths and limitations of this study

► This is the first large-scale nationally representative physical activity (PA) study in 10–11-year- old children across Scotland.

► The study demonstrates the substantial impact of alternative analytical methods on the prevalence of PA levels.

► Study results are comparable with other UK-based accelerometry studies, which provides confidence in published PA levels in similar age groups.

► Non-waterproof waist-mounted accelerometers have known limitations such as their inability to measure water-based activity and to record the acceleration of upper body movement.

## INTRODUCTION

The benefits of an active lifestyle in children and young people are well established, and include the management of overweight and obesity, improved musculoskeletal health and a number of cardiovascular/metabolic benefits (eg, lower blood pressure, cholesterol and blood lipids level).[1] In 2011, a UK-wide consensus among the four chief medical officers (CMOs) advocated that children and young people aged between 5-18 years should accumulate at least 60 min, and up to several hours, of moderate to vigorous intensity physical activity (MVPA) every day[2] and, for the first time, recommended that young people should minimise time spent being sedentary.

National and international evidence on the prevalence of physical activity (PA) comes from a combination of both self-reported (eg, survey/questionnaire/interview) and objective (eg, accelerometry) measurements, with results often discrepant.[3 4] Among Scottish children for instance, the 2015 Scottish Health Survey (SHeS) suggested that 86% of boys and 79% of girls aged between 8 and 10

years meet the current CMO guidelines;[3] however, these figures have been challenged[4] as disproportionately high. Previous UK/European-based objectively measured PA studies have suggested that less than 10% of children and adolescents meet the recommended level of MVPA (eg,[5 6] with some variability across European countries).[7] Self-reported measurements are often criticised for the potential introduction of recall bias, social desirability concerns and general misunderstanding of questions.[8] However, the arguably more robust, objectively measured prevalence estimates also produce equivocal guideline adherence estimates, and studies using accelerometry also face methodological challenges. Previous studies have identified issues regarding epoch length, 'cut-off' points/thresholds used to classify MVPA, and differing measurement devices and models,[4 9] all of which have the potential to influence the underlying construct of MVPA.

A specific concern which transcends both self-reported and objective measurement methods is the approach used to classify participants as either meeting or not meeting the PA guidelines, and the effect of this on the resulting prevalence rates. Current UK guidelines suggest that children (aged between 5-18 years) must meet the threshold of 60 min of MVPA *every day* to be considered 'compliant'. However, some national level surveys, such as the SHeS, assess whether 60 min of MVPA per day is achieved *on average* across 7 days in order to assess prevalence of meeting the guidelines. The UK wide Millennium Cohort Study (MCS), for example found that 53% of 7/8-year-old children met the guidelines using the average approach,[10] whereas, the self-reported health behaviour in school aged children and the parental-proxy reported Health Survey for England found 25% (11-years-old) and 21% (8–10-years-old) of children met the current recommended guidelines using the more stringent 'every day' approach.[11] These are substantial differences that could have serious implications, not least for the evaluation and development of policy. As far as we are aware, only one study has investigated this issue with objectively measured data.[12] In an Estonian sample of children aged 7–13 years (n=472), the authors demonstrated a more than twofold increase in prevalence estimates between an 'every day' guideline approach compared with an average approach (24% vs 52%), although adherence to the daily approach was classified as meeting the guidelines on four out of five weekdays (where no weekend days were included).

An additional issue with regard to guideline adherence is the 60 min threshold. Current evidence suggests that meeting the guidelines has a positive health impact,[13] yet no evidence exists to suggest that this is the definitive threshold at which benefits are gained or lost. In fact, evidence generally suggests that some activity is better than no activity, but 'more is better'.[14] Little evidence exists demonstrating the impact on adherence prevalence rates of altering this threshold by even small margins, and more knowledge regarding this may be of wider value to behaviour change interventions.[12]

Given this apparent inconsistency in the assessment of children's PA guideline adherence, we compared the proportion of Scottish children currently meeting the current CMO-defined PA guidelines when (i) a *minimum* of 60 min of MVPA on each day of valid data was required to be classified as adherent (which we term the *daily* approach); and ii) the *mean* MVPA across valid days was ≥60 min in total (which we term the *average* approach). Additionally, we investigated the impact of altering the 60 min threshold on prevalence rates for both approaches (daily and average), comparing 50, 55, 65, and 70 min. We also describe levels of, and gender/ socioeconomic/seasonal differences in, PA levels in Scottish children.

## METHODS

To examine these questions, we drew on the SPACES (Studying Physical Activity in Children's Environments across Scotland) study, the aim of which was to explore the environmental determinants of PA by conducting a large-scale, nationally representative, accelerometry and global positioning systems (GPS) observational study. The participants involved in SPACES were recruited from the Growing up in Scotland (GUS) study, a nationally representative longitudinal cohort study originating in 2005. As part of the sweep 8 interviews (conducted between September 2014 and February 2015 when the children were aged approximately 10-years-old), parents and children were provided with brief information about the SPACES study and asked if their contact details could be passed on to SPACES staff. From a possible 2402 children, who had participated in GUS sweep 8 interviews, 90% (n=2162) of parents consented to be contacted by us, and we sent study information, registration documents and consent forms by post using the main parent/ carer as primary contact. The data collection for SPACES took place between May 2015 and May 2016.

### Measurement

To assess the frequency, intensity and duration of PA, participants who consented to participate in the data collection were provided with a validated[15 16] accelerometer (ActiGraph GT3X+) and asked to wear the device over eight consecutive days for the waking hours. We included days as valid if they had 10 hours on weekdays, and 8 hours on weekend days.[17] This was to balance having enough data to reliably represent daily activity, and recognise wear time is lower during weekend days (when children may be more likely to spend more time in bed). Children were asked to remove the accelerometer when bathing or during other water-based activities, or during contact sports or activities. Following the measurement protocol set out by the International Physical Activity and Environment Network , we identified that a device was not being worn if there were 60 consecutive minutes of zero acceleration recorded by the device and these

periods were removed from any analyses. From the same recommendations, children who provided at least 5 days including four weekdays and one weekend day were included in the analyses.[18]

## Design
### Physical activity analysis
Proprietary software from the accelerometer manufacturer (ActiLife V. 6.11.9) was set to save PA data in 10 s epochs; PA information was digitised and stored as 'counts'—a unitless representation of acceleration for that period.

The primary 'overall' PA measure was the participant's counts per minute (cpm)—a measure of total PA that integrates all movement recorded through the device over the duration of the device-wearing period (total counts divided by total wear time). These 'counts' were used to calculate time spent sedentary and in each intensity of PA by using an evidence-based threshold classification:[19 20] sedentary (<100 cpm); light (101–2295 cpm); and moderate to vigorous (>2296 cpm). The proportion of children meeting the PA guidelines was classified through two different approaches:

First, in strict agreement with the CMO statement,[2] we calculated the proportion who recorded at least 60 min of MVPA *on each valid day of recording* (the *daily* approach).

Second, we calculated the proportion of children who achieved an *average* of at least 60 min of MVPA across valid days of recording (the *average* approach). To explore the impact of varying the 60-min threshold, we then recalculated these proportions using alternative thresholds (50, 55, 65 and 70 min). These were chosen to reflect what we considered to be slight variations, in both directions, in the duration threshold and would hopefully capture those children who narrowly missed the 60-min threshold while also demonstrating the impact of requiring a further 5 or 10 min to meet the daily guidelines.

## Prevalence estimates
Prevalence estimates were calculated for boys and girls and across a measure of multiple area-level deprivation using ranked scores (grouped into quintiles) from the Scottish Index of Multiple Deprivation (SIMD).[21] This is a well-validated measure that captures the relative social, economic, environmental and health circumstances of local populations. Seasons were classified into a four-level categorical variable based on the astronomical calendar.

## Statistical analysis
Analyses were conducted using STATA V. 13 (STATA Corporation), and accounted for the clustered and stratified survey sample design of the GUS cohort.[22] Sampling weights were applied to allow for non-consent to contact, and non-consent and non-compliance of those invited to take part. This approach maximises the

accuracy of the point estimates and the standard errors from the analyses, therefore reducing the likelihood of Type 1 errors. For this particular analysis, we were more concerned with controlling for the effects of clustering on SE estimation rather than exploring it, and this is why we did not use alternative methods such as multilevel modelling.

Multiple linear regression models, allowing for the survey's design, were conducted on continuous outcome variables (light, MVPA and sedentary time) controlling for number of valid days, mean wear time per day and season of measurement. Logistic regression was used to model the proportion of children meeting the PA guidelines. All covariates within the multiple linear regression analyses were included in the logistic regression models. All models were ran separately for 'sex' and 'SIMD' and the models exploring 'sex' included an interaction term with 'season of measurement'. All pairwise and postestimation tests were Bonferroni corrected to adjust for multiple comparisons.

## RESULTS
From the 2162 GUS children who consented to be contacted, 1096 (51%) children took part in the data collection. From these 1096 participants, 774 children (417 girls; 357 boys) provided at least four weekdays of data and at least 1 day of weekend data. Data from the supplementary file (see online supplementary table 1) compares our weighted sample with that of the GUS-weighted sweep 8 sample to infer representativeness (the GUS-weighted sample being considered to represent the population from which it came). In general, the weighting was successful across all variables. Compared with the entire GUS sweep 8 sample, our weighted sample slightly under-represented those in lowest and highest income bands (<£3999–£9999; >50k), those whose parents were married, those whose mothers were aged under 20 years at the birth of their child, those whose parents had lower level educational grades or equivalent and those who reside in urban areas of Scotland. The sample slightly over-represented those who were cohabiting or single, and in the highest income band (>£50 k).

Adjusted means of derived sedentary and PA outcomes are summarised in table 1. Mean PA levels were 648 cpm, with boys exhibiting higher levels of total activity than girls (this difference just failed to reach significance at the 5% level (P=0.06)). On average, children spent 7.5 hours sedentary per day, with no differences between boys and girls. On average, children spent 73 min per day in MVPA, where boys exhibited significantly higher levels of MVPA than girls (see table 1). There was no significant socioeconomic patterning across any outcome variable, although those in the most deprived quintile did seem to exhibit higher total activity, MVPA, % meeting the *daily* approach guidelines and lower sedentary time than other quintiles of multiple deprivation.

**Table 1** Physical activity and sedentary outcomes (n=774)

| | n (weighted) | Total PA (cpm) | Sedentary all days (hours) | Light all days (hours) | MVPA all days (min) | Daily guideline prevalence | Average guideline prevalence |
|---|---|---|---|---|---|---|---|
| All children Mean (95% CI) | 774 | 648 (627–670) | 7.5 (7.4–7.6) | 4.2 (4.2–4.3) | 72.6 (70.0–75.3) | 11.1% (7.4–14.7) | 68.3% (63.3–73.3) |
| Boys | 359 | 663 (638–688) (P=0.06) | 7.5 (7.3–7.6) | 4.2 (4.1–4.3) | 77.5 (74.1–80.9)* | 12.5% (6.7–18.2) | 75.9% (70.4–81.5) |
| Girls | 415 | 634 (606–661) | 7.6 (7.4–7.7) | 4.3 (4.2–4.4) | 67.7 (64.4–71.0) | 9.8% (5.6–14.0) | 59.5% (63.3–73.3) |
| SIMD Quintile† most deprived | 164 | 668 (607–728) | 7.3 (7.0–7.7) | 4.4 (4.2–4.6) | 75.0 (67.0–83.0) | 19.2% (8.4–30.1) | 68.4% (56.0–80.8) |
| Quintile 2 | 140 | 631 (577–685) | 7.7 (7.4–7.9) | 4.1 (3.9–4.3) | 70.7 (64.7–76.8) | 8.6% (0.4–16.8) | 69.3% (56.6–82.0) |
| Quintile 3 | 143 | 656 (625–688) | 7.5 (7.3–7.6) | 4.3 (4.2–4.4) | 72.8 (68.0–77.5) | 9.5% (4.0–15.0) | 65.8% (56.7–74.9) |
| Quintile 4 | 166 | 645 (607–683) | 7.5 (7.4–7.7) | 4.3 (4.2–4.4) | 71.0 (66.5–75.5) | 9.8% (3.0–16.6) | 66.2% (55.8–76.5) |
| Least deprived | 161 | 635 (626–668) | 7.6 (7.5–7.7) | 4.1 (4.2–4.3) | 71.4 (68.2–74.7) | 7.1% (3.4–10.7) | 65.8% (58.2–73.4) |

Figures are estimated marginal means or adjusted predicted probabilities and significance testing adjusted for season of measurement, mean wear time and number of valid days.

Statistically significant difference between boys and girls, or across SIMD quintiles: *P<0.001

†Significance testing conducted as contrast between base category (most deprived) and all other quintiles

CPM, counts per minute; MVPA, moderate to vigorous physical activity; PA, physical activity; SIMD, Scottish Index of Multiple Deprivation.

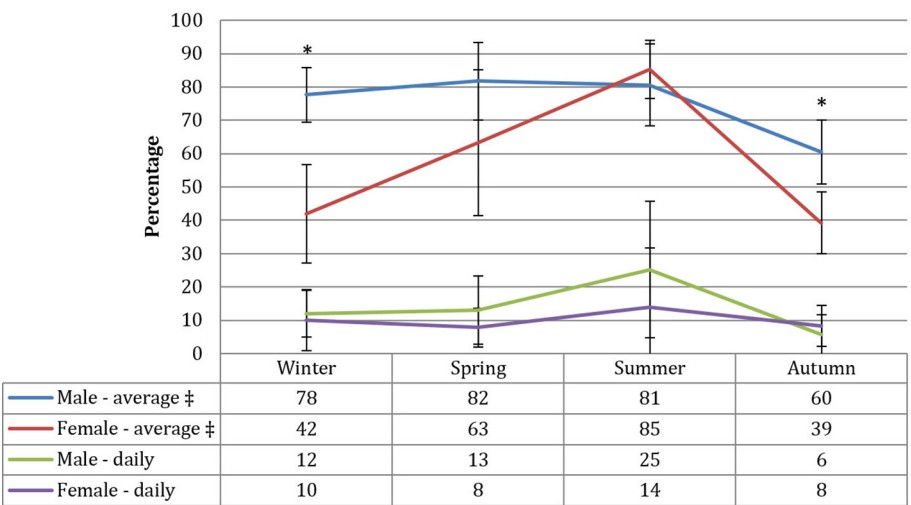

**Figure 1** Physical activity prevalence across season of measurement by sex and measurement approach. *P<0.01; Statistically significant difference between boys and girls within season. ‡ Statistically significant interaction between gender and season, P<0.05. Figures are predicted probabilities and significance testing is adjusted for mean wear time and number of valid days.

### Prevalence estimates—impact of changing analytical approach and adherence criteria

Both approaches produced significantly different estimates (P<0.05). Using the *daily* definition of adherence to assess guideline prevalence, approximately 11% of all children met the current recommendations. There was no significant difference between boys and girls. No statistically significant patterns were found by area deprivation, although a slightly higher proportion of those in the most deprived quintile met the guidelines (see table 1). Although more boys met the guideline across all seasons except during autumn, there was no significant interaction (P=0.7) between gender and season (see figure 1).

Using the *average* approach, 68% of children achieved the recommended level of activity, and significantly more boys met the guidelines than girls (see table 1).

No significant differences were evident between quintiles of deprivation. Contrary to the daily approach, there was a significant interaction between gender and season (P=0.047): a significantly higher proportion of boys met the 60 min daily average in winter and autumn. However, a higher non-significant proportion of girls met the 60 min daily average in summer compared with boys (see figure 1).

Figure 2 illustrates the impact of varying the PA guideline reference point around 60 min. For both approaches, as expected, there was a linear increase in the proportion of children meeting the guidelines as the reference point was lowered from 70 to 50 min. No significant differences existed between boys and girls at any reference point for the daily approach, although the gap widened as the reference point was lowered. There was, however, a consistent significant difference between boys and girls across all

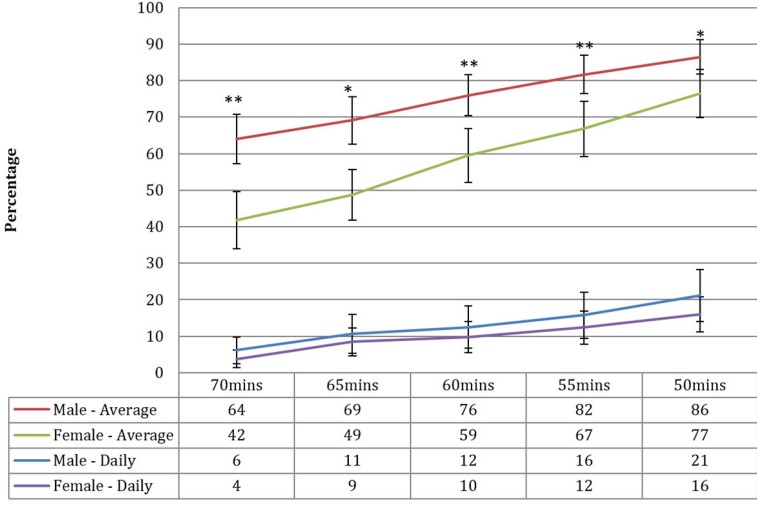

**Figure 2** Physical activity prevalence by sex and threshold classification reference—daily and average approaches. *P<0.01, **P<0.001; statistically significant difference between boys and girls within threshold reference point and measurement approach. Figures are predicted probabilities and significance testing is adjusted for season of measurement, mean wear time, and number of valid days.

levels when using the average approach, although with a tendency for the gap to reduce as the reference point was lowered.

## DISCUSSION

### Statement of principal findings

This was the first representative study of objectively measured PA and sedentary levels among 10–11-year-old children across Scotland. Using the daily measure (which required children to have at least 60 min of MVPA on every valid day of recording), only 11% of Scottish children aged 10–11 years met the recommended guideline. However, this figure substantially increased to 68% when implementing the average measure (which required children to have at least 60 min daily MVPA on average across all valid days). Our results also indicated that reducing or increasing the 60-min threshold by even as little as 5 min can substantially impact the prevalence rates. If the threshold was reduced to 55 min per day, we predict an increase in between 2 and 8 percentage points depending on sex and type of analysis (using an average or daily approach).

### Strengths and limitations

This study was the first to investigate the objectively measured PA levels and guideline prevalence in a large, nationally representative, sample of 10–11-year-old Scottish children. The data reduction protocol followed those of other large studies,[6] allowing accurate comparisons to be made between this study and others, particularly with consideration to the cut points advocated in the literature to classify sedentary behaviour (<100 cpm) and MVPA (>2296 cpm).[6] A further strength of this study is the comparison of two approaches with the interpretation of guideline adherence, and assessment of the subsequent impact on interpretation of PA levels.

The study also had limitations. Waist mounted devices are typically poor at recording the acceleration associated with cycling or upper body dominant activities.[9] The devices were removed for water-based activities and contact sports, so we will have underestimated these activities. Additionally, while our chosen cut points to classify MVPA were evidence-based,[19] other published cut points are available and their use would significantly alter our results.

### Comparison with literature

The PA levels from this study are largely comparable with other large-scale, UK, objectively measured studies of PA in similar age groups (See table 2).

The literature is relatively consistent in showing differences in PA levels between boys and girls and the recent publication by Cooper and colleagues,[6] with over 6000 children aged 9–10 years across seven different countries, supports this. We did not find any statistically significant socioeconomic gradient in activity levels, although there was a tendency for those in the

**Table 2** Descriptive data from large, UK-based, accelerometry studies using similar age groups

| Study name | Setting | Year | Mean age (years) | Sample size | Mean CPM | Cut point for Sed/MVPA | Sedentary (hours) | MVPA (min) | Prevalence of PA (%) |
|---|---|---|---|---|---|---|---|---|---|
| SPACES | Scotland wide | 2015–2016 | 11.1 | 774 | 622 | <100 >2296 | 7.5 | 72 | 11/68.3% |
| MCS | UK (Scotland Sample) | 2008–2009 | 7.5 | 761 | 615 | <100 >2241 | 6.4 | 62 | 52.5% |
| GMS | North East England | 2008–2009 | 9.3 | 405 | 643 | <1100 >3200 | Not reported | 24 | 5.7% |
| SPEEDY | South East England | 2007 | 10.2 | 1862 | 665 | <100 >2000 | 7.6 | 74 | 70.5% |
| ALSPAC | South West England | 2003–2005 | 11.8 | 5595 | 580 | <100 >3600 | 7.2 | 20 | 2.5% |

ALSPAC, avon longitudinal study of parents and children[5]; CPM, counts per minute; GMS, gateshead millennium cohort study[33]; MCS, millennium cohort study[10]; MVPA, moderate to vigorous physical activity; PA, physical activity; SPACES, studying physical activity in children's environments across Scotland; SPEEDY, sport physical activity and eating behaviour: environmental determinants in young people[34].

most deprived areas to engage in higher levels of total activity, MVPA and less time sedentary than other quintiles of deprivation. Again, recent work from the MCS[10] and previous other large-scale objective measured studies[5 23] have shown little socioeconomic patterning at this age. It has been suggested, however, that active travel behaviours may be more prevalent among children from poorer neighbourhoods. They may, for example, be more likely to walk to school compared with children from more affluent neighbourhoods who are more likely to be driven to school.[24] This could explain the slightly higher levels of MVPA among children in more deprived neighbourhoods in our study. Future work with the SPACES dataset will explore this question. The seasonal variation evident in the present study was also consistent with previous work in the UK.[25 26] Atkin and colleagues,[26] for instance, found similar patterns across season but with boys consistently more active than girls when exploring time spent in MVPA. From an intervention, behaviour change and policy perspective, it is vitally important to tailor our approaches to consider the differential effects of season on PA levels, particularly for girls. We plan to explore, using collected GPS data and geographic information systems software, whether specific weather conditions and season influence the probability of missing the 60-min threshold and, therefore, help to explain some of the discordance between the two approaches of guideline adherence.

Our most important finding is that either 11% or 68% of Scottish children meet the current UK guidelines, depending on the approach taken to measurement. This difference could be interpreted as either 'quite poor' or 'relatively good' and would have serious implications no matter which interpretation was taken. The only other study, known to the authors, who have published data using similar approaches, was conducted in Estonian children and young people (6–13-years-old). Mooses and colleagues,[12] using an identical MVPA 'cut point' (ie, the Evenson cut point), identified a greater than twofold increase in guideline prevalence dependent on approach (daily, 24%; average, 52%). However, the authors did not include weekend days and daily adherence was classified as meeting four out five weekdays. The present study required a minimum of four weekdays and one weekend day to be included in analyses, and adherence to the daily approach was based on meeting each and every valid day of wear. Furthermore, the present study was specific to children aged 10–11 years and nationally representative. Therefore, our results are not directly comparable. Previous studies have also identified the impact on prevalence estimates when using different 'cut points' to discern MVPA. The results from work by Griffiths and colleagues,[10] for instance, indicated that the proportions of children meeting the PA guidelines changed from 84.0%/59.4% (boys and girls, respectively) to 13.7%/0.4%, when altering their cut point from 2000 to 3000 cpm. There have been calls for an international consensus process on accelerometry methodology and reporting standards in youth, with a formal review of recommendations every 5 years to evaluate new science.[18] The recent creation of the International Children's Accelerometry Database (ICAD, 6) where standardised methods have reanalysed and reintegrated data from multiple international studies leading to cross-country comparisons is an important step in recognising the need to standardise measurement approaches.

The question remains, however, which approach is more appropriate? Does activity 'every day' matter more, in health benefit terms, than the accumulation over the week? Is 'missing' 1 day out of 1 week problematic in terms of health?[1 27] Furthermore, although the latest evidence demonstrates that meeting the 60 min of MVPA per day has positive health implications, particularly for adiposity and quality of life,[13] there is little evidence on whether individuals whose activity levels approach, but do not reach, 60 min per day are less healthy, and whether narrowly falling short of 'every day' has implications for health outcomes.[12] We know that activity under the prescribed intensity of MVPA has substantial benefits on the maintenance of energy balance, that is, non-exercise activity thermogenesis[28 29]; evidence is growing of the benefits of light intensity activity on other aspects of health, such as the reduced risk of all-cause mortality,[30] and the inverse association with type-2-diabetes risk factors (eg, 2-hour plasma glucose levels) in adults.[31] As far as the authors are aware, no literature exists on the optimal levels, and patterns, of PA intensities combined. Our results also highlight that the proportions of children meeting the UK guidelines vary considerably across 10 min either side of the 60 min threshold. It would be beneficial if studies elsewhere could investigate the health effects of differing patterns of daily MVPA (and sedentary and light PA) and the health-related implications of changing the 60 min threshold. Doing so may result in greater flexibility around guideline adherence, and will provide researchers with improved knowledge when communicating health messages to the public.

## Implications for policy

In 2014, Scotland launched the Active Scotland Outcomes Framework, which set out ambitions for PA and sports in Scotland over the next 10 years. A range of indicators will track progress using SHeS data. Alongside the framework is a key national legacy 10-year programme designed to influence population levels of activity in adults and children.[32] The extent to which these initiatives are viewed as successful in increasing children and young people's PA will depend on the measures used to assess PA and we hope that our findings can inform the understanding of PA prevalence, as well as evaluation methods used to measure the impact of policy change. Griffiths and colleagues[10] were right to state that the clarity of cut points used for MVPA is vital for monitoring of policy objectives around PA but we also demonstrate here that the approach used to assess guideline adherence will

determine the result and, therefore, the evaluation of interventions and policy objectives.

**Acknowledgements** We would like to thank the children from the growing up in Scotland longitudinal birth cohort study for taking part in the research and to members of the Scotcen social research team who assisted along the way.

**Contributors** PM and AE ran the data collection process, and PM led on data processing and data analysis and drafted the article. RM and AE assisted with statistical analysis and interpretation of the data, and RM contributed to the graphical presentation of the results. AE and RM critically revised the article and all authors signed off the final draft.

**Funding** This work was supported by the Medical Research Council [grant number MC_UU_12017/10] and Chief Scientist Office [grant number SPHSU10]; and the Scottish Government [grant number SR/SC 17/04/2012].

**Competing interests** None declared.

**Patient consent** Obtained.

**Ethics approval** Ethics Committee of College of Social Sciences, University of Glasgow (CSS ref: 400140067).

**Provenance and peer review** Not commissioned; externally peer reviewed.

**Data sharing statement** We are committed to maximising the use of SPACES study data to advance knowledge to improve young people's health and welcome proposals for collaborative projects and data sharing. Our data sharing policy follows that of the Medical Research Council and aims to balance making data as widely and freely available as possible with safeguarding the privacy of participants, protecting confidential data, and maintaining the reputation of the study. please contact Prof Anne Ellaway (anne.ellaway@glasgow.ac.uk) for further information.

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
