## [Reviewer comments · BMJ Open]

ARTICLE DETAILS

TITLE (PROVISIONAL)	A comparison of two methods to assess physical activity prevalence in children: an observational study using a nationally representative sample of Scottish 10-11 year old children
AUTHORS	McCrorie, Paul; Mitchell, Richard; Ellaway, Anne

VERSION 1 – REVIEW

REVIEWER	Lisa Price University of Exeter, UK
REVIEW RETURNED	02-Aug-2017

GENERAL COMMENTS	This manuscript examines the difference in PA prevalence rates when different analysis methods are employed. The manuscript presents important findings for those wishing to measure PA and classify children as active or not. It is a well written manuscript that uses a representative sample of children to examine prevalence. There are a number of points that could be addressed: • In the conclusion, for both the abstract and the full text, you note that both methods of prevalence estimates should be employed in future studies. The guidelines state that MVPA should be accumulated on every day, therefore employing the average method will always over estimate those meeting the guidelines. Can you provide justification for why you think both methods should still be employed?• Page 11, second paragraph in methods. The info relating to compliance with wear time and the resultant sample should be in your results section.• Page 8, in the measurement section. Did you ask children to wear the monitors for 10 hours on weekdays and 8 hours on weekend days? Or was this your valid day inclusion criteria? If it was your protocol to ask children to wear it for this amount of time, it could have reduced your sample as longer wear time protocols have demonstrated higher compliance.• Can you justify/ clarify your use of 50 – 70 minutes as your alternate thresholds?• Results: you repeat a lot of information in the text that is presented in the table. You only need to put this in one or the other.
--

REVIEWER	Ryan Donald Burns University of Utah USA
REVIEW RETURNED	10-Aug-2017

GENERAL COMMENTS	Thank you for inviting me to review this manuscript exploring the prevalence of physical activity using different analytical methods in a large cohort of 10-11 year-old children from Scotland. The paper is well written and well analyzed. Although descriptive, this work will add to the knowledge base. There are several instances where clarifications are needed throughout the manuscript. My specific comments are attached. Abstract:  -A clearer purpose statement should be made, specifically the second half of the Objective sentence "...conditional of the analytical decisions made." -Do not start a sentence off with a number (line 26, 45). -Establish that daily approach was statistically different from average approach in this cohort. -The conclusion does not align with the communicated results within the abstract (over-generalize-not based on inferential statistics). Please revise. Introduction:  -Reference needed for line 31. Methods:  -Explain why there is different wear-time criteria on weekdays compared to weekend days. -Please provide a reference for the count cut-points for PA intensity. -Please provide a more thorough explanation for providing weights to account for the clustered data structure. What benefit does this approach have over employing a multi-level mixed-effects model? -If interaction terms were derived for sex, why were separate models run for sex? Please clarify (p. 10, lines 37-42). Discussion:  -Reference needed for page 14 lines 29-32. -PA is often used as a proxy for estimated energy expenditure (EE). Does Scotland have EE guidelines and how may daily or average PA analytical methods affect EE estimates? -How does Scotland's weather patterns reflect the discordance in daily vs average methods (e.g., rainy and cold days may limit outside PA opportunities on any given day)?
---

VERSION 1 – AUTHOR RESPONSE

Reviewer: 1

Reviewer Name

Lisa Price

University of Exeter, UK

Comment: Please leave your comments for the authors below This manuscript examines the difference in PA prevalence rates when different analysis methods are employed. The manuscript presents important findings for those wishing to measure PA and classify children as active or not. It is a well written manuscript that uses a representative sample of children to examine prevalence.

Response: We would like to thank the reviewer for taking their time to read and comment on our paper. Their comments have been integrated into the paper where possible and specific answers can be read below.

Comment: There are a number of points that could be addressed:

- In the conclusion, for both the abstract and the full text, you note that both methods of prevalence estimates should be employed in future studies. The guidelines state that MVPA should be accumulated on every day, therefore employing the average method will always over estimate those meeting the guidelines. Can you provide justification for why you think both methods should still be employed?

Response: The CMO guidelines have been set to reflect the levels of activity where children will experience health benefits. As such, these should be reported. However we also believe we are presenting an incomplete picture if these are the only figures that are presented. The two final sections of the paper tried to present an argument for health, and then an argument for policy (i.e. accurate monitoring of population levels of PA).

There is no definitive evidence regarding duration (i.e. 60 mins) and this number largely reflects intervention studies and observational work that has identified greater health benefits for 60 mins compared to 20, 30, 40 etc. A dose-response relationship does exist and so more is generally better; however, the 60 minute figure was also chosen because it presented a 'guideline' that could foster habit formation. We certainly do not disagree with this but the resulting figures fail to recognise total volume of MVPA or patterns of MVPA – the CMO guideline effectively measures consistency of MVPA at a specified level.

There is (as far as we are aware) no definitive work that has investigated the relationship between differing patterns of activity and health outcomes, where – as we try to state in the paper – someone who misses the 60 minute threshold on 1 day will automatically fail to be recognised as meeting the guidelines, where in fact their activity may be quite high, and subsequent indicators of health outcomes (most likely) positive. We don't have any evidence yet that compares this type of person, with someone who meets the guidelines every day, every second day, someone who misses one day but only by a few minutes, someone who misses one day but falls considerably short etc. If we only focus on meeting the guidelines every day, we are presenting an incomplete picture of this health behaviour.

One figure decidedly states that activity levels are exceptionally poor – when it is doubtful that an individual could meet 60 minutes of MPVA for 365 days per year? The second figure provides a more realistic impression of how active our children are and this may be helpful to numerous stakeholders, including the parents and children themselves, but also to policy makers and the media. Presenting one without the other limits our understanding of the underlying behaviour itself and that's why we think both should be presented.

We have added a sentence in to the abstract and also rewritten the conclusion of the paper to justify its inclusion.

Comment: Page 11, second paragraph in methods. The info relating to compliance with wear time and the resultant sample should be in your results section.

Response: We have taken this paragraph and inserted it into the results section (1st paragraph of results section – see tracked changes).

Comment: Page 8, in the measurement section. Did you ask children to wear the monitors for 10 hours on weekdays and 8 hours on weekend days? Or was this your valid day inclusion criteria? If it was your protocol to ask children to wear it for this amount of time, it could have reduced your sample as longer wear time protocols have demonstrated higher compliance.

Response: Thank you for pointing this out. Children were asked to wear the devices for the waking hours (put on as soon as they woke up and remove when going to bed) for a minimum, where possible, of 12 hours each day, regardless of weekdays or weekend days. It is valid day inclusion criterion that has been presented in the methods. We have amended this sentence to clarify this.

Comment: Can you justify/ clarify your use of 50 – 70 minutes as your alternate thresholds?

Response: We have added a further sentence on page 9, lines 22-31 under the 'physical activity analysis' section to justify our decision to use the alternate threshold durations.

Comment: Results.

you repeat a lot of information in the text that is presented in the table. You only need to put this in one or the other.

Response: Thank you for this comment. We have endeavoured to remove some of the numbers from the text and refer back to Table 1 so as not to repeat. See pages 11 (line 41), 12 (lines 45, 50) and 13 (line 5-6).

Reviewer: 2

Reviewer Name
Ryan Donald Burns

Institution and Country
University of Utah
USA

Please state any competing interests or state 'None declared':
None declared

Comment: Please leave your comments for the authors below Thank you for inviting me to review this manuscript exploring the prevalence of physical activity using different analytical methods in a large cohort of 10-11 year-old children from Scotland. The paper is well written and well analyzed. Although descriptive, this work will add to the knowledge base. There are several instances where clarifications are needed throughout the manuscript. My specific comments are attached.

Response: We would like to thank the reviewer for their time and effort reading our manuscript. We have endeavoured to integrate their comments into the paper where possible and if not we have justified that decision under the specific comments below.

Abstract.

Comment: A clearer purpose statement should be made, specifically the second half of the Objective sentence "...conditional of the analytical decisions made."

Response: We have amended this section to better state our objectives.

Comment; Do not start a sentence off with a number (line 26, 45).

Response; Changed the sentence structure to reflect this comment.

Comment: Establish that daily approach was statistically different from average approach in this cohort.

Response: If we were to formally test whether the population proportion was equal to 11.1% (i.e. the daily approach), we would test the null hypothesis (H_0) that 68.3% (our average approach) was equal ($=$) to 11.1%, with the alternative (H_a : two tailed) being that 68.3% does not equal (\neq) 11.1%. By looking at the 95% CI surrounding our predicted sample proportion (i.e. 68.3) we can see if this overlapped with 11.1. The 95 CI (as stated in Table 1) was LL 63.3 – UL 73.3, and therefore did not overlap, which would suggest that the null can be rejected and the alternative supported - 68.3% does not equal 11.1% at the 5% level.

If we were to do the same thing in reverse to test whether the daily approach was equal to the average approach (i.e. H_0 that 11.1 equals 68.3), our 95 CI surrounding our predicted sample proportion (i.e. 11.1, LL 7.4 – UL 14.7, see Table 1 in manuscript) would suggest that the null could be rejected and the alternative supported (i.e. 11.1 is significantly different from 68.3 at the 5% level).

We have added to our document in the following places:

- Abstract results
- Abstract conclusion
- Results section – under the ‘prevalence estimates’ section

Comment: The conclusion does not align with the communicated results within the abstract (over-generalize-not based on inferential statistics). Please revise.

Response: We have re-written this section.

Introduction:

Comment: Reference needed for line 31.

Response; We have added citations to reference this point.

Methods

Comment: Explain why there is different wear-time criteria on weekdays compared to weekend days.

Response: We have added a sentence to explain why our valid day criteria was different for weekdays and weekend days.

Comment: Please provide a reference for the count cut-points for PA intensity.

Response: We have cited two publications under the ‘Physical activity analysis’ section to provide reference for cut points – one is Evenson et al., (2008), and the other is Trost et al., (2011).

Comment: Please provide a more thorough explanation for providing weights to account for the clustered data structure. What benefit does this approach have over employing a multi-level mixed-effects model?

Response: Thank you for this comment. We have added a sentence and edited this section to further clarify the use of variables to control for survey design effects rather than use MLM techniques.

Comment: If interaction terms were derived for sex, why were separate models run for sex? Please clarify (p. 10, lines 37-42).

Response: We have edited this section. Models were ran separately for sex and SIMD. Season of measurement was included in both models but we also wanted to explore if an interaction existed between sex and season so an interaction term was included in the models for sex only.

Discussion

Comment: Reference needed for page 14 lines 29-32.

Response: We have added a citation to reference this point.

Comment: PA is often used as a proxy for estimated energy expenditure (EE). Does Scotland have EE guidelines and how may daily or average PA analytical methods affect EE estimates?

Response: No energy expenditure guidelines exist for Scotland. We did allude to this with the sentence “As far as the authors are aware, no literature exists on the optimal levels, and patterns, of physical activity intensities combined”. An additional important PA related construct would be physical fitness but we are unsure if by adding in something further, this will slightly confuse our message and hope the above sentence will be seen as addressing this particular point.

Comment: How does Scotland’s weather patterns reflect the discordance in daily vs average methods (e.g., rainy and cold days may limit outside PA opportunities on any given day)?

Response: This is a very interesting question, however, we were unable to collect specific weather variables for individual observations in this study. We plan to obtain that data retrospectively and conduct an exploration of season and weather on PA prevalence.

We have added a sentence to page 16, line 57 through page 17, lines 3-8 within the seasonal variation section of the discussion to reflect this point.